# Mechanosensitive Differentiation of Human iPS Cell-Derived Podocytes

**DOI:** 10.3390/bioengineering11101038

**Published:** 2024-10-17

**Authors:** Yize Zhang, Samira Musah

**Affiliations:** 1Department of Biomedical Engineering, Pratt School of Engineering, Duke University, Durham, NC 27708, USA; 2Center for Biomolecular and Tissue Engineering, Duke University, Durham, NC 27708, USA; 3Division of Nephrology, Department of Medicine, Duke University School of Medicine, Durham, NC 27710, USA; 4Department of Cell Biology, Duke University, Durham, NC 27710, USA; 5Affiliate Faculty of the Developmental and Stem Cell Biology Program, Duke Regeneration Center, Duke MEDx Initiative, Duke University, Durham, NC 27710, USA

**Keywords:** human-induced pluripotent stem cells, stem cell differentiation, podocytes, mechanobiology, biomaterials, matrix elasticity, hydrogel, yes-associated protein, synaptopodin, cytoskeleton

## Abstract

Stem cell fate decisions, including proliferation, differentiation, morphological changes, and viability, are impacted by microenvironmental cues such as physical and biochemical signals. However, the specific impact of matrix elasticity on kidney cell development and function remains less understood due to the lack of models that can closely recapitulate human kidney biology. An established protocol to differentiate podocytes from human-induced pluripotent stem (iPS) cells provides a promising avenue to elucidate the role of matrix elasticity in kidney tissue development and lineage determination. In this study, we synthesized polyacrylamide hydrogels with different stiffnesses and investigated their ability to promote podocyte differentiation and biomolecular characteristics. We found that 3 kPa and 10 kPa hydrogels significantly support the adhesion, differentiation, and viability of podocytes. Differentiating podocytes on a more compliant (0.7 kPa) hydrogel resulted in significant cell loss and detachment. Further investigation of the mechanosensitive proteins yes-associated protein (YAP) and synaptopodin revealed nuanced molecular distinctions in cellular responses to matrix elasticity that may otherwise be overlooked if morphology and cell spreading alone were used as the primary metric for selecting matrices for podocyte differentiation. Specifically, hydrogels with kidney-like rigidities outperformed traditional tissue culture plates at modulating the molecular-level expression of active mechanosensitive proteins critical for podocyte health and function. These findings could guide the development of physiologically relevant platforms for kidney tissue engineering, disease modeling, and mechanistic studies of organ physiology and pathophysiology. Such advances are critical for realizing the full potential of in vitro platforms in accurately predicting human biological responses.

## 1. Introduction

Microenvironmental cues, such as biophysical and biochemical signals, can influence stem cell fate decisions [1,2]. Amongst the physical signals, the elastic moduli of extracellular matrices (ECM) regulate various cellular responses, including migration, proliferation, differentiation, and death [3,4,5,6]. Cells can sense the stiffness of the ECM through transmembrane proteins such as integrins, talins, vinculin, filamin, and α-actinin, which transmit biophysical cues from the ECM to the intracellular environment in the form of contractile forces [7,8]. Such signal transduction enables cells to directly probe and respond to changes in local ECM stiffness. Changes in the contractility of the cell in response to local biophysical cues also modulate and deform the ECM. The interactions between cells and ECM can influence the extent of adaptor protein unfolding, which modulates biochemical responses exhibited by cells, including the choice between survival and death [9,10,11,12,13].

Podocytes are highly specialized visceral epithelial cells with non-proliferative, terminally differentiated, and highly arborized characteristics essential for blood filtration in the kidneys [14,15,16]. Interdigitating podocyte foot processes organize into a zipper-like configuration, forming a molecular sieve around the kidney’s glomerular capillaries. This sieve functions as a porous filter with approximately 40 nm-wide slits that selectively prevent the loss of large blood plasma proteins, such as albumin, while allowing water and small solutes, including toxins, to be filtered and eventually excreted from the body [17,18]. Damage or injury to podocytes increases the leakage of large blood proteins into the filtrate and urine, a condition known as proteinuria, which can lead to the progressive loss of kidney function or chronic kidney disease (CKD) [15,16]. Due to the lack of therapeutic options that can halt CKD, affected patients often progress to end-stage kidney disease (ESKD) and organ failure.

CKD affects more than 10% of the population worldwide and 15% of adults in the United States. Causes of CKD include high blood pressure, diabetes, heart disease, obesity, genetics, infectious diseases (including HIV and COVID-19), and drug toxicities [19,20,21,22,23]. ESKD treatment options are limited to kidney transplantation and dialysis, which are inaccessible to many patients, and neither option is sufficient to fully compensate for the loss of kidney function. Human iPS cell-derived podocytes are a promising source of patient-specific in vitro models for studying kidney biology and disease. They also serve as excellent alternatives to animal models [24,25].

Previous studies have suggested that the differentiation of stem cells and other cellular processes is optimal when carried out on ECMs with stiffnesses characteristic of native tissues [4,26,27,28,29,30,31]. Although the exact stiffness of the healthy human glomerular basement membrane (GBM)—the native ECM for podocytes—is yet to be fully characterized, the stiffness of the rodent glomerulus has been reported to be approximately 2.5 kPa using both micro indentation and capillary micromechanics measurements [32,33], while the stiffness for the renal cortex was shown to range from 2 to 18 kPa as measured by ultrasound elastography [34]. Conditionally immortalized podocytes and renal progenitor cells have been shown to exhibit higher expression of podocyte lineage identification markers, such as podocin and nephrin, when propagated on ECM with a stiffness of 2–12 kPa compared to tissue culture plates (TC plates, which are >10,000 kPa) [35,36,37]. However, the impact of matrix stiffness on human iPS cell-derived podocytes has been less explored. Notably, the widely used standard plastic TC plates are several orders of magnitude stiffer than native tissues, highlighting the need for alternative, physiologically relevant platforms for studying podocyte biology and mechanotransduction.

In this study, we employed polyacrylamide hydrogels to uncover the role of substrate stiffness on podocyte differentiation, gene and protein expression levels, cell survival, and the modulation of mechanosentive targets, including YAP, synaptopodin, and α-actinin-4. We identified nuanced molecular distinctions in podocyte responses to matrix elasticity, uncovering the biophysical regulation of kidney glomerular podocyte development and lineage specification.

## 2. Materials and Methods

### 2.1. Synthesis of Polyacrylamide Hydrogels

#### 2.1.1. Preparation of Reactive Coverslips

All steps for hydrogel synthesis before functionalization should be performed under a fume hood with proper chemical protective equipment. Polyacrylamide hydrogels were synthesized according to our previously published protocol [4], which is a modification of another protocol [28,38]. First, “reactive” coverslips were prepared by treating 18-mm glass coverslips (Thermo Fisher Scientific, Waltham, MA, USA; 12-545-100P) with 0.1 M NaOH solution (Sigma, St. Louis, MO, USA; 567530-500GM) for 3 min; the solution was then removed by aspiration. The coverslips were then covered with 97% 3-aminopropyl-trimethoxysilane (Sigma, St. Louis, MO, USA; 281778) (3-APTMS) for 5 min; following aspiration of the 3-APTMS, the coverslips were washed three times (10 min each) with distilled water. The coverslips were cleared/semi-dried by aspiration and then covered with 0.5% glutaraldehyde solution (Sigma, St. Louis, MO, USA; G7651) for 30 min. The glutaraldehyde solution was removed by aspiration, and the coverslips were washed three times (10 min each) with distilled water. Finally, the coverslips were set aside to air-dry. 

#### 2.1.2. Preparation of Siliconized Coverslips

Siliconized coverslips were prepared by treating 18-mm glass coverslips with 10% SurfaSil siliconizing reagent (Thermo Fisher Scientific, Waltham, MA, USA; TS-42800) in chloroform (Thermo Fisher Scientific, Waltham, MA, USA; BP1145-1) for 15 min. The reagent was removed by aspiration, and the coverslips were washed three times (10 min each) with chloroform. Afterward, the coverslips were washed three times with distilled water (10 min each) and set aside to air-dry. 

#### 2.1.3. Hydrogel Polymerization

The polyacrylamide hydrogels were synthesized using the following solutions: 40% acrylamide aqueous solution (Thermo Fisher Scientific, Waltham, MA, USA; BP1402-1), 2% bis-acrylamide aqueous solution (Thermo Fisher Scientific, Waltham, MA, USA; BP1404-250), 10% ammonium persulfate (APS) solution in water (Sigma, St. Louis, MO, USA; A3678), saturated acrylic acid N-hydroxy-succinimide ester (acrylic-NHS) solution in water (Thermo Fisher Scientific, Waltham, MA, USA; AC400300010), N, N, N′, N′-tetramethylethylenediamine aqueous solution (Thermo Fisher Scientific, Waltham, MA, USA; BP150-20) (TEMED), and distilled water. The stiffness of hydrogels was modulated by adjusting the dilution of bis-acrylamide solution with distilled water in 400 µL of the pre-polymer solution containing 75 µL of acrylamide, 118 µL of acrylic-NHS, 4 µL of APS, and 0.3 µL of TEMED. The hydrogels with stiffness values of 0.7, 3, and 10 kPa were synthesized using 10, 60, and 120 µL of bis-acrylamide solution, respectively. After briefly vortexing the pre-polymer solution, 140 µL of the solution was pipetted onto a “reactive” coverslip. A siliconized coverslip was gently placed on top of the pipetted pre-polymer solution to form a sandwich-like structure of the pre-polymer solution between the two coverslips. Polymerization was allowed to proceed for 10 min. The siliconized coverslip was gently removed after polymerization, and the hydrogels (bound to the “reactive” coverslips) were transferred to a 12-well plate.

#### 2.1.4. Functionalization of Hydrogels and Preparation for Cell Culture

The transferred hydrogels were washed three times (10 min each wash) with Dulbecco’s phosphate-buffered saline solution (Thermo Fisher Scientific, Waltham, MA, USA; 14190144) (DPBS) and then incubated for 2 h with 1 mL of a solution containing 5 mM N-(2-aminoethyl)maleimide trifluoroacetate salt (Sigma, St. Louis, MO, USA; 56951). The pH of the salt solution was adjusted to 7.5 with 0.1 M NaOH solution. The treated hydrogels were washed three times (10 min each wash) with sterile DPBS. Afterward, the hydrogels were sterilized by exposing them to 1 mL of 70% ethanol solution (VWR, Radnor, PA, USA; 89125-172) for 5 min. The sterile hydrogels were washed three times (5 min each wash) with sterile DPBS. To promote cell adhesion, the hydrogels were functionalized by overnight incubation with recombinant laminin-511 (iMatrix-511) at 0.5 mg/mL in sterile distilled water (Takara, Kusatsu, Shiga, Japan; T303) on a rocker. The functionalized hydrogels were transferred to a sterile 12-well plate and washed three times (10 min each wash) with sterile DPBS and one time with sterile water and then stored at 4 °C until further use.

### 2.2. Human iPS Cell Culture

All cell lines were obtained under appropriate material transfer agreements, approved by all involved institutional review boards, tested for mycoplasma contamination using a MycoAlert Mycoplasma Detection kit (Lonza, Morristown, NJ, USA; LT07-318), and cultured by following established methods. Briefly, the human iPS cell line PGP1 (Personal Genome Project) was propagated on TC plates (VWR, Radnor, PA, USA; 10062-892) that were coated with Matrigel (BD Biosciences, Franklin Lakes, NJ, USA; BD354277) and using mTeSR1 medium (Stem Cell Technologies, Vancouver, BC, Canada; 85850). The human iPS cells were incubated at 37 °C in 5% CO_2_ and passaged after 4–5 days by treatment with StemPro accutase (Thermo Fisher Scientific, Waltham, MA, USA; A1110501).

### 2.3. Differentiation of Human iPS Cells into Intermediate Mesoderm (IM)

According to the previously established protocol for human iPS cells in podocyte induction [27,39,40], human iPS cells were first dissociated from Matrigel-coated plates by treatment with StemPro accutase and centrifuged twice at 200 RCF for 5 min each in DMEM/F12. The cells were plated on laminin-511-E8 (Takara, Kusatsu, Shiga, Japan; T303) (iMatrix-511)-coated plates with a mesoderm differentiation medium consisting of DMEM/F12 with GlutaMax (Thermo Fisher Scientific, Waltham, MA, USA; 10565018) supplemented with 100 ng mL^−1^ activin A (Thermo Fisher Scientific, Waltham, MA, USA; PHC9564), 3 μM CHIR99021 (Stemgent, Cambridge, MA, USA; 04-0004), 10 μM Y27632 (TOCRIS, Minneapolis, MN, USA; 1254), and 1 × B27 serum-free supplement (Thermo Fisher Scientific, Waltham, MA, USA; 12587010). After 2 days of differentiation, the cells were cultured for a minimum of additional 14 days with an IM induction medium containing DMEM/F12 with GlutaMax and supplemented with 100 ng mL^−1^ BMP7 (Thermo Fisher Scientific, Waltham, MA, USA; PHC9543), 3 μM CHIR99021, and 1 × B27 serum-free supplement.

### 2.4. Substrate-Induced Differentiation of IM Cells into Podocytes

To induce the podocyte phenotypes, the IM cells were dissociated by treatment with 0.05% trypsin-EDTA, split at a 1:4 density onto freshly prepared laminin-511-E8-coated plates/hydrogels with various stiffnesses, and fed daily for either 3 or 5 days with either (1) a podocyte induction medium consisting of DMEM/F12 with GlutaMax supplemented with 100 ng mL—1BMP7, 100 ng mL^−1^ activin A, 50 ng mL^−1^ VEGF (Thermo Fisher Scientific, Waltham, MA, USA; PHC9393), 3 μM CHIR99021, 1 × B27 serum-free supplement, and 0.1 μM all-trans retinoic acid (Stem Cell Technologies, Vancouver, BC, Canada; 72264) for a 5 day duration or (2) the podocyte induction medium for 3 days followed by culture boost medium for 2 days (Cell Systems, Kirkland, WA, USA; 4N0-500) with 1% Pen Strep. 

### 2.5. Cell Viability Assay

Podocytes cultured on the hydrogels or TC plates were incubated for 2 h (at 37 °C with 5% CO_2_) with CCK-8 (Sigma, St. Louis, MO, USA; 96992) reagent diluted in CultureBoost medium at a ratio of 1:100. Then, 100 µL of the mixture from each well was then transferred to a 96-well clear flat bottom plate (Greiner, Kremsmünster, Austria; 655094) for absorbance reading at the 450 nm wavelength using a plate reader (VWR, Radnor, PA, USA; 75886-128). CCK-8 data were collected from two samples per condition per replicate experiment, and five independent experiments were performed. For each time point, the relative podocytes viability was obtained by normalizing absorbance readings of podocytes on each condition to culture boost medium alone (blank condition), and the relative cell viability values were plotted using GraphPad Prism software version 10.

### 2.6. Immunostaining and Microscopy Analysis

Cells were fixed by incubation with 4% formaldehyde (28908, Thermo Fisher Scientific) in DPBS for 20 min. The fixed cells were permeabilized by incubation with permeabilization buffer (0.125% TritonX-100 (VWR, Radnor, PA, USA; 0694-1L) in DPBS) for 5 min. The permeabilized cells were blocked by incubation with 1% bovine serum albumin (Sigma, St. Louis, MO, USA; A9418) and 0.125% TritonX-100 in DPBS for 30 min at room temperature. The blocked cells were washed three times with permeabilization buffer and then incubated with primary antibodies at the manufacturer’s recommended dilutions in permeabilization buffer overnight at 4 °C. Afterward, the samples were washed three times with permeabilization buffer and then incubated with secondary antibodies at 1:1000 dilution in permeabilization buffer for 1 h at room temperature. Primary antibodies used for immunostaining analysis included α-actinin-4 (Abcam, Boston, MA, USA; ab108198), nephrin (American Research Products, Waltham, MA, USA; 03-GP-N2), protein kinase C (PKC) λ/ι (Santa Cruz Biotechnology, Dallas, TX, USA; sc-376344), podocin (Abcam, Boston, MA, USA; ab50339), synaptopodin (Santa Cruz Biotechnology, Dallas, TX, USA; sc-515842), YAP (Santa Cruz Biotechnology, Dallas, TX, USA; sc-101199), and TEAD1 (Cell Signaling Technology, Danvers, MA, USA; 12292). Secondary antibodies used for immunostaining included Alexa Fluor 488 (Life Technologies, Waltham, MA, USA; A21202) and Alexa Fluor 594 (Life Technologies, Waltham, MA, USA; A21203) conjugated antibodies. To visualize the nuclei, the cells were counterstained with 4′, 6-diamidino-2-phenylindole (DAPI, Invitrogen, Waltham, MA, USA; D1306) at a dilution of 1:1000 in distilled water for 5 min at room temperature. For staining F-actin filaments, the cells were incubated with Alexa Fluor 594-conjugated phalloidin (Invitrogen, Carlsbad, CA, USA; A12381) at a dilution of 1:200 in DPBS for 20 min at room temperature. Microscopy images of the cells were captured using an EVOS M7000 microscope (Thermo Fisher Scientific, Waltham, MA, USA; AMF7000). All images were analyzed using Fiji ImageJ, which included the quantification of protein level expressions and cytoskeletal arrangement. Cytoskeleton arrangements were analyzed using ImageJ’s Orientation J plug, which enabled identification of the orientation and isotropy of actin filaments in single cells. Nuclear-to-cytoplasmic expression of YAP was quantified using Cell Profiler version 4.2.6 and GraphPad Prism version 10.

### 2.7. Western Blot Analysis

Whole-cell extracts were acquired by lysing the cells with RIPA buffer (50 mM Tris-HCl, 150 mM NaCl, 1% NP-40, 0.5% sodium deoxycholate, and 0.1% SDS), followed by fractionation by SDS–PAGE, and transferred to a nitrocellulose membrane using a transfer apparatus (Bio-RAD, Hercules, CA, USA; 1704150) according to the manufacturer’s protocols. The membranes were blocked by treatment with 5% non-fat milk in TBST (50 mM Tris-HCl, 150 mM NaCl, 0.1% Tween-20) followed by incubation with mouse anti-GAPDH (Millipore, Burlington, MA, USA; MAB374), rabbit anti-α-actinin-4 (Abcam, Boston, MA, USA; ab108198), guinea pig anti-nephrin (American Research Products, Waltham, MA, USA; 03-GP-N2), rabbit anti-podocin (Abcam, Boston, MA, USA; ab50339), mouse anti-synaptopodin (Santa Cruz Biotechnology, Dallas, TX, USA; sc-515842), mouse anti-YAP (Santa Cruz Biotechnology, Dallas, TX, USA; sc-101199), and rabbit anti-TEAD1 (Cell Signaling Technology, Danvers, MA, USA; 12292) antibodies. A horseradish-peroxidase-conjugated goat anti-rabbit, anti-mouse, or anti-guinea pig antibody was then added, and the membranes were developed according to the vendor’s protocol (Bio-RAD, Hercules, CA, USA). Quantification of the band intensity was performed using ImageJ software version 1.54e.

### 2.8. Statistical Analysis

Data were analyzed by *t*-tests and one-way analysis of variance (ANOVA) Tukey’s multiple comparisons test for differences of means at a 95% confidence level. Differences were considered significant at * *p* < 0.05, ** *p* < 0.01, *** *p* < 0.005, and **** *p* < 0.0001. All statistical analyses were performed using GraphPad Prism 10.

## 3. Results and Discussion

### 3.1. Stiffness-Dependent Adhesion and Differentiation of Human iPS Cell-Derived Podocytes

The stiffness of glomerular tissue is estimated to be ~2.5 kPa [32,33] (Figure 1a). Although this value was determined using rodent samples and there are limited reports on human tissues, it is estimated that the human glomerular tissues have similar mechanical properties. Specifically, shear wave elastography measurements on human volunteers indicated an average stiffness of 5.4 kPa for the healthy kidney [41]. We aimed to recapitulate the physiologically relevant stiffness using polyacrylamide hydrogels. Biomedical applications of polyacrylamide hydrogels have been described previously—these include their use in therapeutic delivery, tissue engineering, and cosmetics [42,43,44]. We utilized previously established protocols [4,28,38] and chemo-selective reactions [45] to synthesize polyacrylamide hydrogels and functionalize [46] them with desired cell-adhesive epitopes and promote podocyte differentiation (Figure 1b). In addition, we validated the swelling capacity at equilibrium by measuring the thickness of our 3 kPa and 10 kPa hydrogels (Appendix A) and found the thickness to be at a comparable level to that in the original Musah et al. 2012 study [4].

To differentiate human iPS cells into mature podocytes, we utilized the established differentiation medium [27,39,40] in a 17-day protocol. We first differentiated human iPS cells into mesoderm by culturing dissociated human iPS cells in a mesoderm induction medium (described in Section 2). Then, we replaced mesoderm induction medium with IM induction medium for an additional 14 days to generate IM cells (Figure 1c). The IM cells were then seeded on the hydrogels or control TC plates and induced to podocytes, as described above. 

Softening and stiffening of the GBM occur in some kidney diseases, including those arising from collagen deficiency and Alport Syndrome [47,48]. We examined the effects of matrix elasticity on podocyte differentiation using hydrogels with varying stiffnesses (0.7, 3, and 10 kPa). We also compared results from podocytes differentiated on standard TC plates. After one day of culture, cells adhered to each of the hydrogels; however, there were more cells attached to the 3 kPa and 10 kPa hydrogels than the more compliant 0.7 kPa hydrogels. After 3 days of differentiation, we observed clustering of podocytes on the 3 kPa and 10 kPa hydrogels but not on 0.7 kPa hydrogels. By day 5 of podocyte differentiation, the 3 kPa and 10 kPa hydrogels exhibited stable podocyte-like cell populations (Figure 2a), but significantly fewer viable cells could be found on the 0.7 kPa hydrogels at a very low level (Figure 2b). Often, on the 0.7 kPa hydrogels, the cells failed to complete the induction period due to significant cell loss and detachment; we therefore excluded the 0.7 kPa hydrogels from subsequent analyses.

### 3.2. Hydrogels with Tissue-like Stiffness Promote Podocyte Viability, Differentiation, and Expression of Lineage Identification Markers

To examine how human iPS cell-derived podocytes respond to different matrix elasticities, we initially quantified cell viability after cell culture and differentiation on the hydrogels. We observed that podocytes differentiated on 3 kPa and 10 kPa hydrogels exhibited similar levels of viability (Figure 2b, Appendix A). Phalloidin staining for F-actin and phase contrast imaging were employed to characterize cell morphology and cytoskeletal arrangement in the podocytes on the hydrogels that supported cell viability (Figure 2c). We also compared the cells on the hydrogels with those differentiated on TC plates. We observed that F-actin fibers were more aligned in podocytes differentiated on 10 kPa hydrogels compared to 3 kPa hydrogel. Intriguingly, on 10 kPa hydrogel, aligned F-actin was found in many of the cell bodies and extended well into the foot processes. In contrast, the F-actin cytoskeleton in podocytes differentiated on the 3 kPa hydrogels was relatively less aligned or elongated and condensed mainly to the cell bodies (Figure 2c). Furthermore, by quantifying actin filament alignment using the OrientationJ plugin in the Fiji ImageJ software (version 1.54e), we found that podocytes on 3 kPa hydrogels exhibited statistically significant lower actin skeleton coherency than podocytes on 10 kPa hydrogels (Figure 2d). Taken together, these findings suggest that matrix elasticity modulates differential cytoskeletal arrangement during podocyte differentiation. Specifically, podocytes differentiated on the 3 kPa substrate produce less organized and condensed actin networks and decreased cytoskeletal coherency than the podocytes differentiated on the 10 kPa matrix. Disorganization of the cytoskeleton is associated with podocytopathies and podocyte foot process effacement (FPE) [49,50]. Given the observation that 0.7 kPa hydrogels cause significant cell loss and detachment, it is conceivable that this compliant hydrogel failed to stabilize the cell’s cytoskeletal structure necessary for sustained adhesion, viability, and differentiation. Thus, although the 3 and 10 kPa hydrogels exhibit different levels of F-actin fiber arrangement and coherency, they supported stable cell adhesion and differentiation, underscoring the sensitivity of human iPS cell-derived podocytes to matrix stiffness.

We examined the role of matrix stiffness in podocyte lineage identification marker expression. In vivo, development of the slit diaphragm is regulated by multiple podocyte proteins, including nephrin and podocin. Nephrin helps to maintain the slit diaphragm structure and facilitates selective filtration of molecules across the glomerular filtration barrier. Nephrin also plays a role in cell-cell signaling, injury response, and tissue physiology [51,52]. Podocin is an integral raft-associated protein that traffics nephrin to the cell membrane and interacts with other proteins to form and maintain the slit diaphragm [53,54]. Proper expression of nephrin is required for podocyte lineage determination, cell maturation or specialization, and function. Consequently, prenatal knockdown of nephrin induces significant levels of proteinuria, as experimentally confirmed in mice [55], and knockout of podocin in prenatal mouse models leads to severe proteinurea in the antenatal period and subsequent renal failure and death within a few days after birth [56].

Immunofluorescence (Figure 2e) and Western blot (Figure 2f) analyses revealed that podocytes differentiated on both 3 kPa and 10 kPa hydrogels have robust expression levels of podocin and nephrin (Figure 2e–h), comparable to podocytes differentiated on TC plates. Our results on cytoskeletal structure and lineage identification protein (nephrin and podocin) expression suggest the presence of a mechanosensitive pathway that maintains the podocyte molecular phenotype despite the observed differences in actin coherency. These findings also suggest that matrices with physiologically relevant stiffnesses can promote nephrin and podocin expression in differentiated podocytes in vitro despite the significant mechanical difference from traditional tissue culture vessels.

### 3.3. Expression and Subcellular Localization of YAP in Differentiating Podocytes

The transcriptional coactivator YAP is a component of the Hippo signaling pathway, and it shuttles between the cell nucleus and cytoplasm to regulate gene transcription [57]. On stiff substrates, YAP is generally known to localize to the nucleus, and on soft substrates, it is retained mostly in the cytoplasm in many cell types [58]. In specialized and functional podocytes in vivo and in vitro, YAP is predominantly localized to the nucleus, and its expression and nuclear localization are crucial for podocyte survival and function [59,60,61]. We examined the mechanosensitive roles of YAP and its interacting factor TEAD1 in human iPS cell-derived podocyte development. We performed immunofluorescence (Figure 3a) and Western blot (Figure 3e) analyses of YAP and TEAD1 expression in podocytes differentiated on the hydrogels. Intriguingly, total YAP levels were significantly higher in podocytes differentiated on the 3 kPa hydrogels when compared to the relatively stiffer 10 kPa hydrogels (Figure 3e,g), which was also supported by our immunofluorescence results (Figure 3b,c). Furthermore, we observed a higher level of nuclear (transcriptionally active) YAP in podocytes differentiated on the 3 kPa substrate than the 10 kPa substrate, while the cytoplasmic YAP level was similar between the two hydrogels (Figure 3b,c). Further analysis of the nuclear/cytoplasmic YAP ratio indicated that podocytes differentiated on the 3 kPa substrate exhibit more nuclear expression of YAP compared to podocytes differentiated on the 10 kPa substrate (Figure 3d, Appendix A). Intriguingly, there were no significant differences in TEAD1 expression levels between podocytes differentiated on the 3 kPa and 10 kPa hydrogels (Figure 3e,f).

We speculate that the higher YAP nuclear localization and higher YAP expression levels observed in the podocytes differentiated on the 3 kPa hydrogels might be associated with the moderate cytoskeleton coherency shown in Figure 2 and described above, allowing the cells to more readily conform to morphological changes that may be critical for foot process development and organization, as well as tissue formation. While this could be the subject of future studies, it correlated with the YAP upregulation observed in vivo following kidney injury linked to the disruption of podocyte cytoskeletal integrity [59,60,61]. Because YAP nuclear localization is crucial for podocyte survival, function, as well as cytoskeleton structure [62,63,64], the higher levels of YAP (both active and total YAP) observed in podocytes propagated on the 3 kPa hydrogels suggest that this substrate has optimal rigidity or biophysical properties (similar to kidney tissue, Figure 1a) for podocytes to reorganize their cytoskeletal structure, remain adherent, and express key lineage identification markers and mechanosensitive transcriptional network.

### 3.4. Hydrogels with Kidney-Tissue-like Stiffness Enhance Synaptopodin Expression in Differentiated Podocytes

The actin-binding protein α-actinin-4 is associated with podocyte biological processes such as actin filament assembly, focal adhesion maturation, enhanced cell adhesion, and cell migration [65,66,67]. Genetic mutations in α-actinin-4 have been implicated in focal segmental glomerulosclerosis [68]. Synaptopodin is another actin-binding protein that exists in podocyte foot processes, and the absence of synaptopodin has resulted in increased susceptibility to Adriamycin-induced nephropathy in mice [69]. To explore whether changes in podocyte actin binding protein expression levels could underlie the differences in cytoskeleton organization observed in podocytes differentiated on the 3 kPa hydrogel, we performed immunofluorescence (Figure 4a) and Western blot (Figure 4b) analyses of α-actinin-4 and synaptopodin. We found similar levels of synaptopodin and α-actinin-4 expression in podocytes differentiated on the 3 kPa and 10 kPa hydrogels (Figure 4b–d). Interestingly, the expression level of synaptopodin in podocytes differentiated on 3 kPa hydrogels was significantly higher than the podocytes differentiated on TC plates (Figure 4c), further supporting the evidence that hydrogels with kidney-tissue-like stiffness facilitate the differentiation of podocytes and molecular-level characteristics better than traditional TC plates. Thus, it is conceivable that the 3 kPa hydrogel (which is closer to the native stiffness of the kidney glomerulus) might be a viable in vitro system for recapitulating and understanding podocyte biomechanics, mechanotransduction, and the role of synaptopodin in kidney health and disease.

## 4. Summary and Conclusions

In summary, this study demonstrates that human iPS cell-derived podocytes differentiated on hydrogels with different stiffnesses show distinct morphological and molecular profiles. Specifically, extremely soft substrates (0.7 kPa) failed to support podocyte adhesion and differentiation, while intermediate or relatively stiffer hydrogels (3 and 10 kPa) significantly promoted the adhesion and differentiation of human iPS cell-derived podocytes. We uncovered molecular-level differences in the podocytes differentiated on the hydrogels; podocytes differentiated on the 3 kPa hydrogels express significantly higher levels of YAP compared to 10 kPa hydrogels. Additionally, YAP localization studies showed that podocytes differentiated on the 3 kPa hydrogels have higher nuclear to cytoplasmic localization of the YAP protein when compared to podocytes differentiated on 10 kPa hydrogels. 

Given the importance of YAP nuclear localization and transcriptional activation in podocyte biology, health, and stress response [62,63], we speculate that the higher levels of total YAP and increased YAP nuclear localization observed in podocytes differentiated on the 3 kPa hydrogels may be linked to the dynamic nature of the moderate cytoskeleton coherency (compared to more rigid substrates) and help maintain proper cell phenotype, morphology, and response to microenvironmental cues. The transcriptional regulation of YAP on the hydrogel may play a role in regulating the expression levels of podocyte lineage markers, including nephrin, podocin, and synaptopodin, as explored in this study.

Notably, the expression levels of synaptopodin were significantly higher in podocytes differentiated on the 3 kPa hydrogels than podocytes differentiated on TC plates (Figure 4c). Synaptopodin integrates signals from tyrosine and serine/threonine kinase and regulates the balance between RhoA and Rac1 activity, allowing the maintenance of stress fibers and podocyte structures [70]. Previous reports have shown that primary podocytes in ex vivo culture exhibit decreased synaptopodin expression and altered cytoskeletal organization compared to their freshly isolated counterparts [71]. The hydrogels used in this study—which are cell culture matrices designed to closely mimic the biophysical properties of tissues—can induce higher expression levels of podocyte slit diaphragm-related proteins than traditional tissue culture environments, highlighting the importance of stiffness in podocyte differentiation and kidney tissue engineering. Nuclear exclusion (or increased cytoplasmic or transcriptionally inactive levels) of YAP can exacerbate the progression of podocyte loss in Adriamycin-induced focal segmental glomerulosclerosis [60,61]. Our laboratory has previously demonstrated that YAP-target genes, such as Cyr61 and CTGF, and YAP-interacting factor TEAD1 are down-regulated in human iPS cell-derived podocytes during Adriamycin-induced injury [63]. This study further illustrates the importance of YAP in podocyte viability and phenotype as the cells adapt to various substrate stiffnesses. Our findings underscore the role of YAP in podocyte biology and stability.

This work has implications for future studies on podocyte injury response and repair, especially in kidney diseases that result from changes in the biomechanical or mechanosensitive properties of the GBM, as indicated in Alport syndrome [47,72] and diabetic nephropathy [73]. By modulating the expression levels of YAP genetically or epigenetically, it might be possible to ameliorate the biophysical stress response and potentially protect podocytes from injury and CKD progression. In addition, we suspect that the excessive softness of the 0.7 kPa hydrogel might hinder the cytoskeletal polymerization, assembly, and focal adhesion formation necessary for stable podocyte adhesion to the hydrogels. Future studies could explore the amount and distribution of focal adhesions in podocytes propagated on matrices with varying stiffness and their response to glomerular injury stimuli. Finally, understanding how podocytes respond to substrate stiffness can inform strategies to develop biomaterial platforms for kidney tissue engineering. Future studies could also examine the molecular mechanisms of podocyte response to biophysical cues and the interplay between YAP and the podocyte markers, including podocin and nephrin, in health and disease.

## Figures and Tables

**Figure 1 bioengineering-11-01038-f001:**
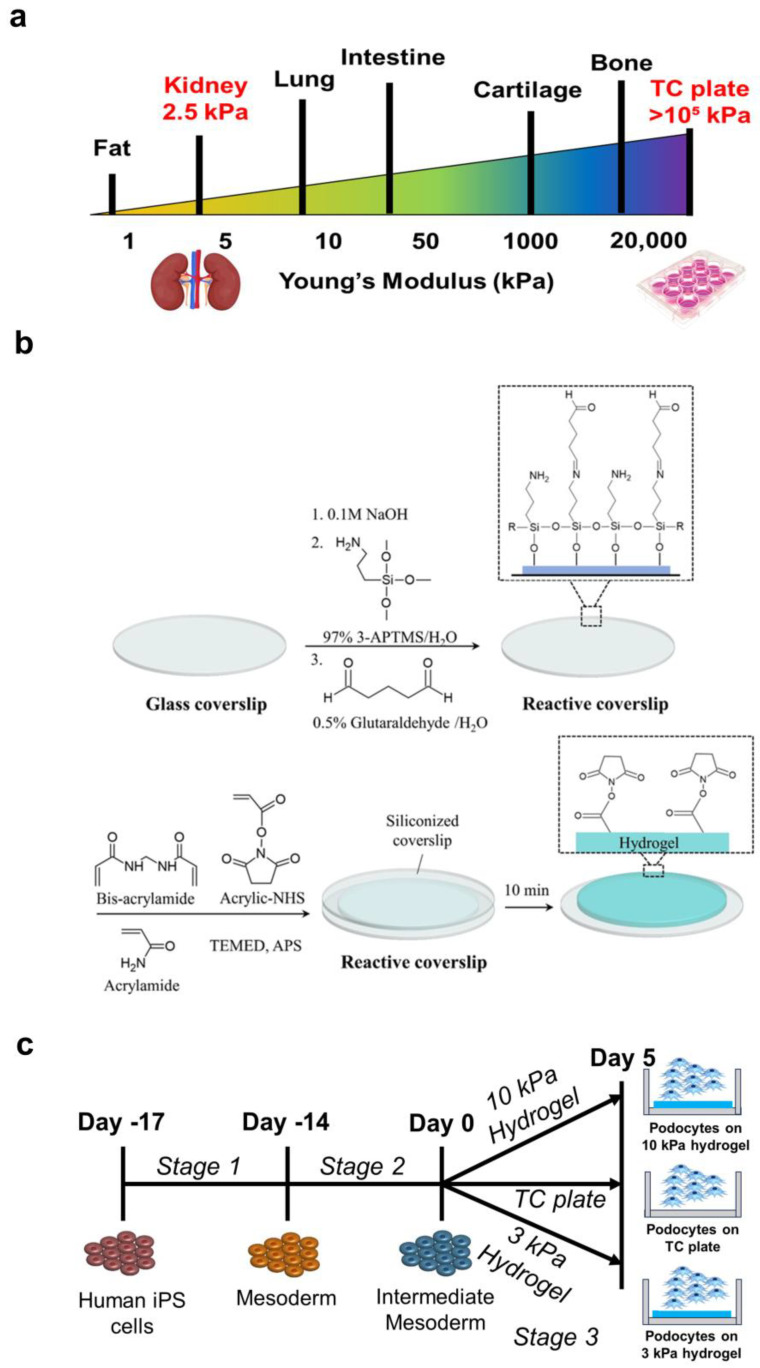
Overview of the procedure for hydrogel synthesis and podocyte differentiation. (**a**) Schematic illustration showing the relative stiffness of various organs, including the kidney, as indicated by the elastic modulus. (**b**) (1) Hydrogel preparation starting with glass coverslip etching using sodium hydroxide and aminosalinization with 97% 3-APTMS, and glutaraldehyde treatment serving as an agent to facilitate hydrogel immobilization, thus creating a reactive coverslip. (2) Acrylamide monomers were cross-linked with bisacrylamide using free radical polymerization, initiated and catalyzed by the addition of APS and TEMED. Acrylic-NHS was added to the prepolymer mixture for subsequent functionalization of the hydrogel. Siliconized coverslips, which are non-adhesive to the polyacrylamide hydrogel, were applied to help the polymer mixture spread into uniform, thick, circular-shaped layers and were removed after the polymerization was complete. The hydrogels become fully polymerized within 10 min. (3) Addition of N-(2-aminoethyl) maleimide trifluoroacetate salt facilitates laminin functionalization onto the hydrogels for cell attachment. (**c**) Human iPS cells were differentiated into mesoderm, IM, and then podocyte lineages in a four-stage, 21-day process to obtain podocytes.

**Figure 2 bioengineering-11-01038-f002:**
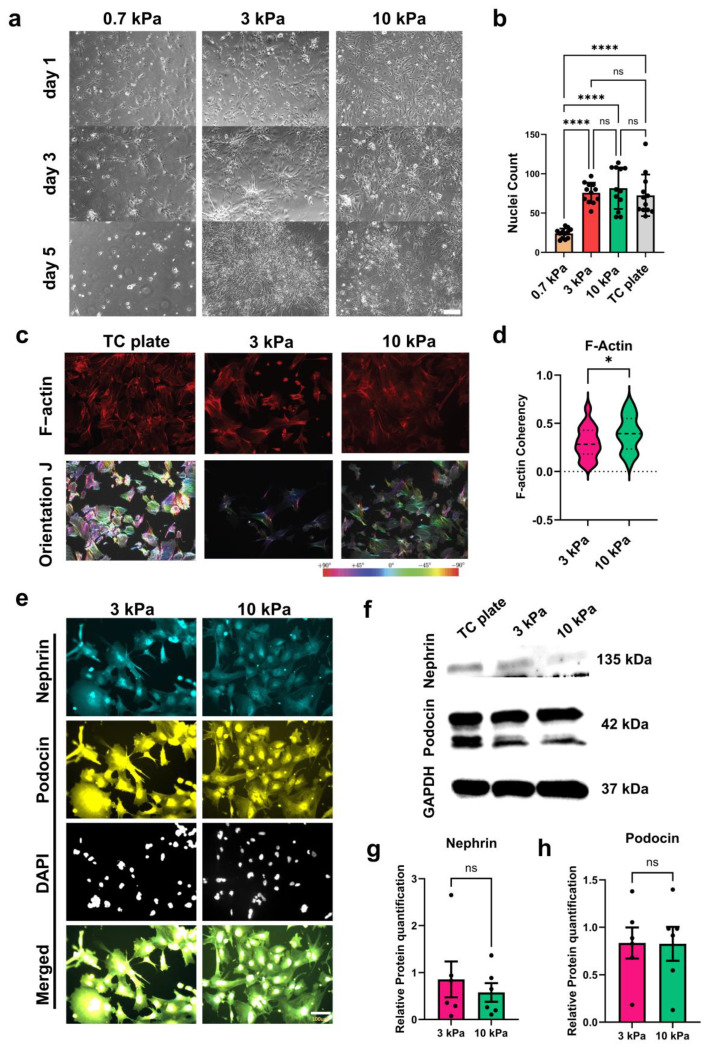
Adhesion and differentiation of human iPS cell-derived podocytes on hydrogels. (**a**) Representative phase contrast microscopy images showing cell adhesion profiles during differentiation from IM to podocytes; scale bar, 1 mm. (**b**) Nuclei (DAPI) count for podocytes cultured on hydrogels of different rigidities (N = three independent experiments, n = nine). (**c**) Representative fluorescence microscopy images of podocytes after 5 days of differentiation on TC plate (~10,000 kPa, Control) compared to the 3 kPa and 10 kPa hydrogels. Samples were stained with phalloidin (red) and analyzed in the Fiji ImageJ OrientationJ plugin (rainbow colors, bottom panel) and the OrientationJ quantification of F-actin fiber coherency is shown in (**d**). (**e**) Representative immunofluorescence images of podocytes differentiated on 3 kPa and 10 kPa hydrogels; nephrin (cyan), podocin (yellow), and DAPI (light gray); scale bar, 100 µm. (**f**) Representative immunoblot showing expression levels of the podocyte markers podocin and nephrin. (**g**) Western blot quantification for podocin; N = 3, n = 6. (**h**) Western blot quantification for nephrin; N = 3, n = 6 Data are expressed as mean ± s.e.m. (ns, not significant, * *p* < 0.05, **** *p* < 0.0001).

**Figure 3 bioengineering-11-01038-f003:**
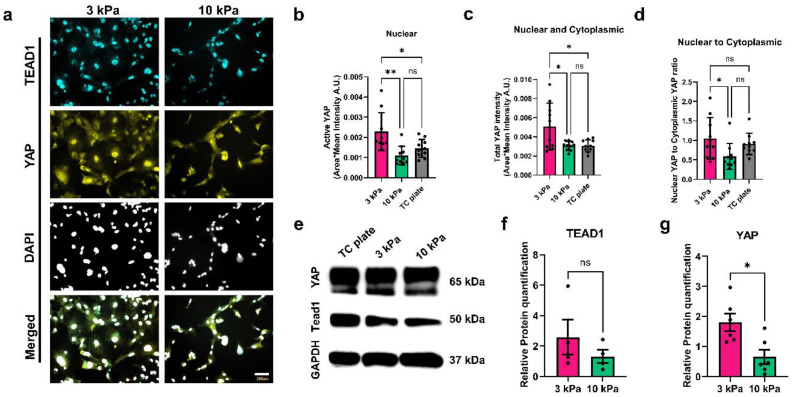
The role of hydrogel stiffness on TEAD1 and YAP expression in human iPS cell-derived podocytes. (**a**) Representative immunofluorescence images of podocytes differentiated on 3 kPa and 10 kPa hydrogels and immunostained for TEAD1 (cyan), YAP (yellow), and counterstained with DAPI (light gray); scale bar, 100 µm. (**b**) Active nuclear YAP and (**c**) total YAP levels, and (**d**) comparative ratios of nuclear to cytoplasmic YAP in cell culture on each hydrogel. (N = 2, n = 10). (**e**) Representative immunoblot showing expression levels of YAP and TEAD1. (**f**) Quantification of TEAD1 (N = 2, n = 4) and (**g**) quantification for YAP (N = 3, n = 6). Data are expressed as mean ± s.e.m. (ns, not significant, * *p* < 0.05, ** *p* < 0.01).

**Figure 4 bioengineering-11-01038-f004:**
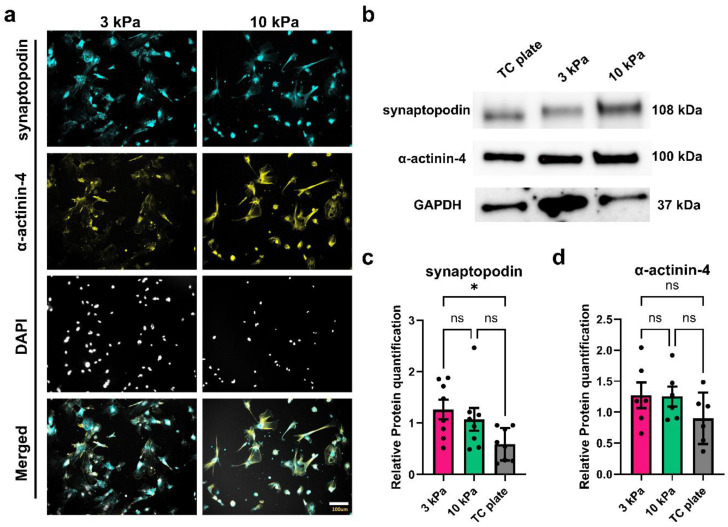
Expression levels of synaptopodin and α-actinin-4 in podocytes differentiated on hydrogels compared to TC plate. (**a**) Representative immunofluorescence images of podocytes differentiated on 3 kPa and 10 kPa hydrogels and immunostained for synaptopodin (cyan) and α-actinin-4 (yellow), and counterstained with DAPI (light gray); scale bar, 100 µm. (**b**) Representative immunoblot showing expression levels of synaptopodin and α-actinin-4. (**c**) Western blot quantification for synaptopodin (N = 2, n = 4) and (**d**) α-actinin-4. (N = 3, n = 6). Data are shown as mean ± s.e.m. (ns, not significant, * *p* < 0.05).

## Data Availability

The data that support the findings of this study are available from the corresponding author upon a reasonable request.

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
