# Peer review of "Mechanosensitive Differentiation of Human iPS Cell-Derived Podocytes"

_bioengineering, 2024, doi:10.3390/bioengineering11101038_

Round 1

Reviewer 1 Report

Comments and Suggestions for Authors

The manuscript entitled “Mechanosensitive differentiation of human iPS cell-derived podocytes” by Zhang et. al, investigates how the matrix stiffness impacts the differentiation of the human iPS towards podocytes. Multiple characteristics of the differentiation process were carefully studied including F-actin orientation, viability, and lineage-maker expression. This work provides insight into kidney tissue engineering, particularly the importance of mechanotransduction during the formation of organoids, and of broad interest. I recommend the manuscript be published after the issue below is addressed.

1.       There are typos in the author affiliation, please check.

2.       Please also check the accuracy of the chemical structure in Fig 1b. Aldehyde reacts with amine and may result in imine rather than amide.

3.       In Fig1b, I would suggest ensuring the surface chemical structure by IR spectrum and confirming the protein adhesion.

4.       In Fig 2e, f,g, both the qualitative and quantitative analysis of podocin and nephrin levels will need a negative control of undifferentiated cells to set the basis of stiffness-induced expression of those lineage markers.

Author Response

We thank the reviewers for their positive feedback about our work. In the revised version of our manuscript, we addressed the reviewer’s comments and suggestions as indicated below:

Reviewer #1

The manuscript entitled "Mechanosensitive differentiation of human iPS cell-derived podocytes" by Zhang et. al, investigates how the matrix stiffness impacts the differentiation of the human iPS towards podocytes. Multiple characteristics of the differentiation process were carefully studied including F-actin orientation, viability, and lineage-maker expression. This work provides insight into kidney tissue engineering, particularly the importance of mechanotransduction during the formation of organoids, and of broad interest. I recommend the manuscript be published after the issue below is addressed.

Comment 1: There are typos in the author affiliation, please check.

Response 1: Thank you for the comments. After double-checking, we confirm that the author affiliation is accurate. The confusion might have arisen from the spelling of the Duke MEDx initiative, but we confirm that this is the correct name for the program. Therefore, we have not made any changes regarding this comment.

Comment 2: Please also check the accuracy of the chemical structure in Fig 1 b. Aldehyde reacts with amine and may result in imine rather than amide.

Response 2: Thank you so much for pointing that out. We agree with your comment. There was a mistake in the chemical structure drawn on the reactive coverslip, and we have updated the chemical structure in Fig 1b.

Comment 3: In Fig1b, I would suggest ensuring the surface chemical structure by IR spectrum and confirming the protein adhesion.

Response 3: Thank you for the comment. We agree that it is important to check for surface chemical structure and protein adhesion using IR spectroscopy. Although we have not performed such an experiment in this paper, we followed the same well-established protocol for polyacrylamide hydrogel preparation as described in Musah et al., 2012. The surface peptide immobilization was thoroughly validated in that paper using using multiple strategies including peptide conjugation and imaging techniques. Furthermore, the hydrogel is inert, so podocyte attachment and growth on the substrate requires an adhesive epitope such as the laminin-coating employed in this study. Therefore, showing that podocytes have good viability and growth on 3 kPa and 10 kPa hydrogels is an indication of proper surface chemical structure and protein adhesion.

Comment 4: In Fig 2e, f,g, both the qualitative and quantitative analysis of podocin and nephrin levels will need a negative control of undifferentiated cells to set the basis of stiffness-induced expression of those lineage markers.

Response 4: Thank you for the keen observation. Indeed, the mechanical properties of a material can influence cell fate for various stem cell types. However, we believe that our human iPS cell-induced podocytes (based on the protocol developed by Musah et al., 2017, published in Nature Biomedical Engineering) exhibit robust expression of podocyte markers nephrin and podocin, along with distinct foot-like process morphology. In contrast, undifferentiated cells at earlier stages (iPS cells, mesoderm, and intermediate mesoderm) show very limited expression of these markers. Additionally, when cultured in an organ-on-a-chip system, these podocytes developed a functional filtration barrier in conjunction with endothelial cells (Mou et al., 2024, Science Advances).

Therefore, it is relatively easy to distinguish between undifferentiated cells and podocytes based on morphology alone when cultured on hydrogel. We can confirm that during the podocyte induction period, the cells gradually transition from an intermediate mesoderm-like morphology to a podocyte-like morphology.

Moreover, undifferentiated iPS cells would not survive in podocyte induction media. Thus, even if we wanted to study the effects of mechanical cues alone on hydrogel, it would introduce additional confounding variables to the experiment whenever an irrelevant cell type is used.

Reviewer 2 Report

Comments and Suggestions for Authors

The manuscript "Mechanosensitive differentiation of human iPS cell-derived podocytes" is interesting and suitable for bioengineering. The introduction includes sufficient data from the literature, and the conclusions are appropriate. The figures and tables are suggestive, conveying the information effectively. The authors need to make some changes and clarifications:

1. Please add in the abstract the composition of the hydrogel and which polymers it contains.
2. Abbreviations are not recommended in the Keywords section. Please change them.
3. In Section 2.1, I recommend the authors add other subsections for a more adequate visualization of the data.
4. Please add a scale of 1 mm to Figures 2e, 3a, and 4a.
5. Representative images of the hydrogel compositions are not highlighted in the article.
6. The stability of the hydrogel samples in physiological environments and the swelling capacity at equilibrium are missing from this study. They represent an important property in terms of the investigation with hyrogel samples for adhesion, viability, and biocompatibility studies.

Author Response

We thank the reviewers for their positive feedback about our work. In the revised version of our manuscript, we addressed the reviewer’s comments and suggestions as indicated below:

Reviewer #2

The manuscript "Mechanosensitive differentiation of human iPS cell-derived podocytes" is interesting and suitable for bioengineering. The introduction includes sufficient data from the literature, and the conclusions are appropriate. The figures and tables are suggestive, conveying the information effectively. The authors need to make some changes and clarifications:

Comment 1: Please add in the abstract the composition of the hydrogel and which polymers it contains.

Response 1: Thank you for pointing that out. We have updated the abstract to include a description of the hydrogel polymer as requested.

Comment 2: Abbreviations are not recommended in the Keywords section. Please change them.

Response 2:  We have replaced the abbreviation (YAP) in the keywords section with its full name: Yes-associated protein.

Comment 3: In Section 2.1, I recommend the authors add other subsections for a more adequate visualization of the data.

Response 3: Thank you for the suggestion. We have separated each step of the hydrogel synthesis process in Section 2.1 into its own section or paragraph to improve the clarity and visualization of the data.

Comment 4: Please add a scale of 1 mm to Figures 2e, 3a, and 4a.

Response 4: We have updated the scale bars in Figures 2e, 3a, and 4a, moving them to the bottom right of each panel and making them clearer for viewers to see.

Comment 5: Representative images of the hydrogel compositions are not highlighted in the article.

Response 5: Thank you for the comment. The representative images of the hydrogel (Figure 1b) are referenced at the end of the first paragraph in the Results and Discussion section. The composition of the hydrogel is discussed in detail in the Materials and Methods section.

Comment 6: The stability of the hydrogel samples in physiological environments and the swelling capacity at equilibrium are missing from this study. They represent an important property in terms of the investigation with hydrogel samples for adhesion, viability, and biocompatibility studies.

Response 6: Thank you for your keen observation. We agree with your point that the stability of the hydrogel and its swelling capacity at equilibrium are important for cell adhesion, viability, and biocompatibility. We would like to first point out that the swelling capacity at equilibrium was well established in the Musah et al., 2012 paper, which we follow exactly for hydrogel preparation. In that original paper, the hydrogel thickness was measured to be 150 µm. To address your comment specifically for this manuscript, we measured the thickness of the hydrogel and found it consistent with this previous measurement. Interestingly, we found that even the hydrogels that had been stored in PBS for 10 months had similar thickness ( 10 kPa hydrogel was approximately 143 µm thick, and the 3 kPa hydrogel was approximately 146 µm thick). In the revised version of this manuscript, we included these new data in Supplementary Figure 4. Together, these data demonstrate the stability of the hydrogel even after 10 months of storage. Additionally, the similar thicknesses between this paper and the Musah 2012 paper further confirm that the swelling capacity at equilibrium for these hydrogels is consistent. Lastly, the covalent crosslinking nature of polyacrylamide hydrogels guarantees their stability under cell culture conditions.